# Cognitive Impairment in Convalescent COVID-19 Patients Undergoing Multidisciplinary Rehabilitation: The Association with the Clinical and Functional Status

**DOI:** 10.3390/healthcare10030480

**Published:** 2022-03-04

**Authors:** Pasquale Moretta, Pasquale Ambrosino, Anna Lanzillo, Laura Marcuccio, Salvatore Fuschillo, Antimo Papa, Gabriella Santangelo, Luigi Trojano, Mauro Maniscalco

**Affiliations:** 1Istituti Clinici Scientifici Maugeri IRCCS, Neurological Rehabilitation Unit of Telese Terme Institute, 82037 Telese Terme, Italy; anna.lanzillo@icsmaugeri.it (A.L.); laura.marcuccio@icsmaugeri.it (L.M.); 2Istituti Clinici Scientifici Maugeri IRCCS, Cardiac Rehabilitation Unit of Telese Terme Institute, 82037 Telese Terme, Italy; antimo.papa@icsmaugeri.it; 3Istituti Clinici Scientifici Maugeri IRCCS, Pulmonary Rehabilitation Unit of Telese Terme Institute, 82037 Telese Terme, Italy; salvatore.fuschillo@icsmaugeri.it (S.F.); mauro.maniscalco@icsmaugeri.it (M.M.); 4Department of Psychology, Università della Campania Luigi Vanvitelli, 81100 Caserta, Italy; gabriella.santangelo@unicampania.it (G.S.); luigi.trojano@unicampania.it (L.T.)

**Keywords:** COVID-19, cognitive impairment, rehabilitation, exercise, outcome, disability, occupational medicine

## Abstract

Background. Cognitive impairment has been reported in the aftermath of severe acute respiratory syndrome due to coronavirus 2 (SARS-CoV-2) infection. We investigated the possible association between cognitive impairment and the main clinical and functional status variables in a cohort of convalescent COVID-19 patients without premorbid diseases potentially affecting cognition. Methods. We consecutively screened for inclusion of convalescent COVID-19 patients referring to a post-acute care facility for pulmonary rehabilitation. All the enrolled patients were assessed for cognitive functions. We also investigated features of psychological distress (anxiety, depression, symptoms of posttraumatic stress disorder and quality of life) and cardiac and pulmonary functional status. Results. The 63 enrolled patients (mean age 59.82 ± 10.78, male gender = 47) showed a high frequency of depressive symptoms (76.2%) and anxiety (55.5%), and a high prevalence of symptoms of posttraumatic stress disorder (PTSD, 44.4%). About half of the total sample showed reduced cognitive efficiency (RCE, 44.4%) in the domains of spatial and verbal long-term memory and executive functions. Patients with RCE more frequently showed alteration of blood pressure (BP) circadian rhythm (*p* = 0.01), higher levels of D-Dimer (*p* = 0.03), had experienced a severe illness (*p* = 0.02), had longer disease duration (*p* = 0.04), more clinically relevant symptoms of PTSD (*p* = 0.02), more frequent cognitive complaints (*p* = 0.002), higher anxiety scores (*p* = 0.01) and lower quality of life (*p* = 0.02) than patients with normal cognitive efficiency. Conclusions. Our findings indicated a possible association between the RCE after COVID-19 and some cardiological variables, including some indirect measures of a residual autonomic disorder, such as the presence of an altered BP circadian rhythm. Future research studies with large samples are needed to provide valid conclusions.

## 1. Introduction

A growing body of evidence suggests the presence of cognitive impairment in the aftermath of severe acute respiratory syndrome due to coronavirus 2 (SARS-CoV-2) infection. A number of neurological manifestations (e.g., myalgia, headache, dizziness, hyposmia, hypogeusia, ataxia-encephalitis, stroke, seizures, sleep problems) are considered both a direct effect of the virus infection on the central nervous system and the consequence of immune-mediated reactions during coronavirus disease 2019 (COVID-19) [1,2,3]. After recovery from the acute illness, many sequelae have been described [4,5,6,7,8,9,10,11]. In the psychological and behavioral domains, a recent large-scale study [5] reported that the prevalence of long-lasting symptoms can be as high as 75% among hospitalized patients, and encompass persistent fatigue, muscle weakness, depression, and non-restorative sleep 6 months after recovery from COVID-19 [6,7]. Moreover, COVID-19 survivors experience a range of symptoms of psychological distress, depression, anxiety, frequent nightmares and post-traumatic stress disorder (PTSD) [8,9,10]. Clinically relevant depression and anxiety were detected in about 30–40% of convalescent patients [2], and one-quarter of them had persisting symptoms even after 6 months [4], regardless of the need for hospitalization [10]. Notably, among hospitalized patients, PTSD was observed in approximately 30%, with a dramatic impact on quality of life [9]. Several studies have also investigated the occurrence of cognitive dysfunctions after COVID-19 by means of screening tools or computerized batteries, showing that cognitive domains, such as memory, attention and speed processing, were mainly impaired [11], both in the post-acute phase [9,12] and up to 6 months after remission [4]. Cognitive impairment is related to difficulties in day-to-day activities and working, with a great functional and emotional impact on patients’ and their families’ quality of life [4]. Unfortunately, until now the mechanisms underlying the neuropsychological deficits after SARS-CoV-2 infection have been poorly investigated; in fact, it is not clear if cognitive deficits may be ascribed to a direct impact of the virus on the brain [13], to indirect factors such as hypoxia and use of mechanical ventilation [14], or both [15].

The effect of the inflammatory response [7] or the psychological outcomes after the hospitalization experience have also been called into question [11].

Since COVID-19 can be considered as a systemic illness, it increases the risk of several non-pulmonary complications such as acute myocardial injury, renal failure or thromboembolic events [2].

The presence of autonomic alteration may affect cognitive functioning, as reflected in the dysregulation of blood pressure (BP) rhythms. This aspect could play an important role in the development of cognitive impairments after COVID-19 since a correlation between hypertension and cognitive functioning has been demonstrated [16,17]. Several observational studies showed that high BP, associated with increased inflammation and oxidative stress, functional and structural vascular changes, and vascular dysregulation, can trigger cerebral small vessel disease, stroke, reduction in brain volumes and ultimately dementia [18].

Unfortunately, this issue has not been studied yet in patients affected with COVID-19. In general, most studies addressing cognitive functions in COVID-19 patients have been performed on small samples [19,20,21], used only cognitive screening tools [9,10,12], and did not correlate the neuropsychological alterations with the clinical and functional status of convalescent COVID-19 patients [15,20,22]. Moreover, no study allowed to exclude the impact of premorbid conditions on the cognitive dysfunctions documented in the post-acute phase [11]. Thus, we aimed at investigating the possible association between cognitive impairment and the main clinical and functional variables in a cohort of convalescent COVID-19 patients without premorbid diseases potentially affecting cognitive performance.

## 2. Materials and Methods

### 2.1. Participants

In the current observational study, we screened a consecutive series of patients admitted to the Rehabilitation Unit of Telese Terme with consequences of severe to moderate pneumonia, according to World Health Organization (WHO) criteria (WHO reference number: 451 WHO/2019nCoV/clinical/2020.5). All patients were admitted to our Rehabilitation Unit within 2 months from negativizing of swab test. Exclusion criteria were: age lower than 18 or higher than 65 years; lack of or poor formal education (primary school not completed); past or current psychiatric or neurologic diseases; presence of cardiovascular or respiratory conditions potentially impacting cognitive performance, including hypertension, diabetes, chronic heart failure, chronic obstructive pulmonary disease, and asthma. All included patients underwent a pulmonary rehabilitation (PR) program with daily sessions (six sessions/week), including physical exercise training, dietary and psychological counseling, in line with the recommendations by the American Thoracic Society/European Respiratory Society [23]. Physical exercise training was the cornerstone of the program and included exercises to strengthen groups of muscles in the upper and lower extremities, treadmill walking and stationary cycling. Participation was supervised and monitored by physiotherapists.

All participants provided their written informed consent. The study was approved by the competent Ethics Committee (Istituto Nazionale Tumori Fondazione Pascale, Naples, Italy, with reference number ICS 11/20), and was performed according to the principles of the Helsinki Declaration. The study followed the Strengthening the Reporting of Observational Studies in Epidemiology (STROBE) reporting guidelines [24].

### 2.2. Main Study Procedures

The study was performed at patients’ admission to the Rehabilitation Unit. Upon study entry, we collected demographic and clinical data. Clinical interviews and medical records were used to assess whether patients suffered from chronic premorbid cardiovascular, respiratory and neurological diseases. We also considered the premorbid drug regimen.

We performed pulmonary function tests using automated equipment (Vmax^®^ Encore, Vyasis Healthcare, Milan, Italy) according to the American Thoracic Society/European Respiratory Society (ATS/ERS) guidelines [25]. We measured forced vital capacity (FVC), forced expiratory volume in 1 s (FEV1), and the FEV1/FVC ratio. Using a gas analyzer (Instrumentation Laboratory, GEM premier 3000, Milan, Italy), we measured arterial blood gas [partial pressure of oxygen (PaO2) and partial pressure of carbon dioxide tension (PaCO2)] and expressed the pressure in mmHg according to the manufacturer’s recommendations.

Using a blood pressure 24-h Holter monitor, we measured circadian blood pressure variations. We computed daytime and nighttime BP as the mean value during daytime and nighttime, respectively. Per our directions, participants carried out their normal activities during the monitoring period, with the only exception of keeping their arms still at the time of each BP reading, and recorded their activities during each measurement. Based on the nocturnal fall in BP, we classified participants into two groups: non-dippers (nocturnal BP fall < 10%) and dippers (nocturnal BP fall ≥ 10%) [26].

We took a venous blood sample from all participants in fasting conditions for measuring C-reactive protein (CRP) and D-Dimer levels.

#### 2.2.1. Psychological Distress and Quality of Life Assessment

All patients completed the following self-report questionnaires:(1)Hospital Anxiety and Depression Scale (HADS), consisting of 14 items, seven for the anxiety subscale (HADS-A), and seven for the depression subscale (HADS-D). Each item is scored on a scale ranging from 0 (no symptom) to 3 (severe symptom). Scores ranging from 8–10 indicate doubtful cases, while scores ≥ 11 indicate clinically relevant cases [27]. A cut-off score ≥ 8 can be considered optimal for both sensitivity and specificity for the diagnosis of clinically relevant anxiety and depression [28];(2)State-Trait Anxiety Inventory-Form Y2 (STAI-Y2) [29], a questionnaire assessing the tendency to react to environmental stimuli with a high level of trait anxiety; it is composed of 20 items to be rated on a 1–4 scale, with higher scores meaning higher levels of anxiety; the cut-off score for the presence of relevant anxiety symptoms is 40;(3)Impact of Event Scale–Revised (IES-R) [30], a 22-item self-report measure assessing subjective distress caused by traumatic events. Items correspond to 14 of the 17 DSM-IV symptoms of PTSD. Respondents were asked to identify a specific stressful life event and then indicate how much they were distressed or bothered during the past seven days by each “difficulty” listed.

Items are rated on a 5-point scale ranging from 0 (“not at all”) to 4 (“extremely”). The IES-R yields a total score (ranging from 0 to 88). The cut-off of 33 indicates clinically relevant symptoms of PTSD [31].

(4)EuroQOL (EQ-5D) [32], evaluating the quality of life of patients with different pathologies according to a continuum of five dimensions (mobility, care person, habitual activities, pain or discomfort, anxiety or depression) and a Visual Analog Scale (VAS) on the perception of one’s state of health on a 20-cm visual analogue scale. Respondents are asked to indicate how they rate their own health state by drawing a line from an anchor box to the point on the VAS which best represents their own health on that day (range 0–100).

#### 2.2.2. Neuropsychological Assessment

We selected five neuropsychological tests to focus the assessment on attention, executive functioning, and long-term memory on the basis of previous studies supporting the relationship of these cognitive domains with functional outcome [16] and with inflammation, vascular processes, hypoxia, and depression and anxiety symptoms [33,34,35]. Our battery also included two screening tools for assessing global cognitive efficiency and frontal functions.

The battery included the following tests:(1)Verbal Fluency Test (FAS) [36,37], requiring to orally produce as many words as possible beginning with the letters F, A, and S within 1 min for each phonetic cue; successful performance requires executive control, selective attention, set-shifting, internal response generation, and self-monitoring;(2)Trail Making Test (TMT) [38,39], assessing attention and executive functions, in which participants are required to connect in ascending order numbers (Part A) or numbers and letters alternately (Part B); performance is expressed as the time (seconds) needed to complete the task; the B-A score (obtained by subtracting Part A time from Part B time) is thought to be related to executive control;(3)Rey auditory Verbal learning task [40,41], assessing long-term verbal memory. Participants are required to learn a 15-word list repeated five times; performance is assessed by counting the number of words recalled after each presentation (immediate recall), and after a filled 15-min interval (delayed recall);(4)Corsi block-tapping test [42,43], assessing visuospatial immediate memory. Participants are required to tap the sequence of blocks tapped by the examiner on a wooden tablet on which nine cubes are irregularly placed. The maximum number of blocks tapped in the correct order is about 5–6 for healthy individuals;(5)Supra-span learning on Corsi’s Test [42], assessing spatial long-term memory. A sequence of eight blocks on the same tablet as before is presented repeatedly up to a maximum of 18 times until participants reach the learning criterion (three consecutive exact repetitions); the total score is computed on the basis of block correctly tapped in the 18 trials;(6)Frontal Assessment Battery (FAB) [44], assessing the following functions related to frontal lobes: (1) conceptualization and abstract reasoning (similarities test); (2) mental flexibility (verbal fluency test); (3) motor programming and executive control of action (Luria motor sequences); (4) resistance to interference (conflicting instructions); (5) inhibitory control (go–no go test); and (6) environmental autonomy (prehension behavior). The cut-off score of FAB is 12/18.(7)Montreal Cognitive Assessment (MoCA) [45], a screening test assessing global cognitive functioning and the following cognitive domains: short-term memory recall task (score ranging 0–5), visuospatial abilities (score ranging from 0–4), executive functions (score ranging from 0–4), attention, concentration, and working memory (score ranging from 0–5), language (score ranging from 0–5), abstract reasoning (score ranging from 0–2), orientation to time and place (score ranging from 0–5). Higher scores indicate better cognitive functioning. The cut-off score for this test is 26/30.

#### 2.2.3. Timeline of the Psychological and Neuropsychological Assessment

As part of a routine evaluation, all convalescent COVID-19 patients admitted to our Rehabilitation Unit completed the neuropsychological assessment that was performed in fixed order during two sessions, at least two days apart, within one week from admission. In the first session, the participants completed MoCA, Rey auditory Verbal learning task, and TMT. In the second session, they completed: FAB, supra-span learning on the Corsi test, and FAS. Moreover, eligible patients were requested to complete the psychological questionnaires (HADS, STAI, IES-R) in paper form at their own ease, within one week after they had received the material. When individuals returned back the filled questionnaires, they were encouraged to discuss any doubt or difficulty in comprehending or in responding to any item.

During the debriefing, all patients were asked to refer if they experienced cognitive difficulties and, in the affirmative case, they were invited to specify which. Patients that referred cognitive difficulties were classified as having cognitive complaints.

The neuropsychological assessment was always performed by an experienced neuropsychologist, in a standard setting in the morning (between 9:30 and 11:30 a.m.), before physiotherapy sessions. The patients and the examiner wore a surgical mask as personal protective equipment.

### 2.3. Statistical Analysis

Continuous data were expressed as mean ± standard deviation (SD). Categorical variables were summarized as relative frequencies.

As for the neuropsychological tests, we compared the scores achieved by each participant with normative values, thus assessing the prevalence and clinical relevance of cognitive impairment. In most Italian normative studies, raw scores are adjusted as a function of single individuals’ age, education and sex (as appropriate), and then converted into a non-parametric five-point ordinal scale (Equivalent Scores, ES) [42]. Within this approach 0 is a score equal to or outside the outer tolerance limit (5%) of normative distribution (i.e., a ‘pathological’ score); 4 is equal or higher than the median value of the normative sample; 1, 2, and 3 are obtained by dividing into three equal parts the area of distribution between 0 and 4; an ES of 1 thus means a ‘borderline’ score, i.e., a score in the low range of the normative sample. On this basis, we identified patients with reduced cognitive efficiency (RCE) as those who achieved pathological scores (ES = 0) on at least two neuropsychological tests when compared to age- and education-matched healthy individuals, and those with at least one pathological score (ES = 0) in one test and one or more borderline score (ES = 1) in other tests. This criterion for clinical classification of cognitive performances was based on the probability to find ‘pathological’ scores (ES = 0) within a neuropsychological battery as a function of the number of tests [46].

We evaluated whether the main clinical variables, as well as the scores on psychological questionnaires, differed according to the presence/absence of RCE by means of non-parametric Mann–Whitney U test, with alpha level set at *p* < 0.002 according to Bonferroni correction for multiple comparisons (number of comparisons = 23).

We also assessed the differences in the prevalence of hypoxia (i.e., PaO2 < 80% predicted), reduced lung function (i.e., FEV1 < 80% predicted), dipping pattern, self-reported cognitive complaints and clinically relevant symptoms of PTSD between patients with RCE and patients with normal cognitive efficiency (NCE) by means of non-parametric Chi-square with a two-tailed alpha level set at 0.05.

We also analyzed the internal consistency of psychological questionnaires (HADS, STAI-Y2 and IES-R), by computing Cronbach’s alpha. According to current guidelines, we considered “acceptable” an alpha value of 0.70, above 0.80 as “good” and above 0.90 as “excellent” [47].

Statistical analysis was carried out using Statistical Package for the Social Sciences (SPSS), version 24 (IBM Corp., Armonk, NY, USA).

### 2.4. Sample Size

Based on the results of a previous meta-analysis on the community prevalence of mild cognitive impairment [48], a sample of 51 convalescent COVID-19 patients was considered necessary to detect an expected pre-defined doubled prevalence with a power of 80% and a level of significance of 5%. To account for possible dropouts, we planned to screen at least 160 patients for eligibility.

## 3. Results

As reported in Appendix A, we screened 193 patients. A total of 65 patients entered the study. Of these, 2 (3.1%) patients withdrew from the study before completing projects requirements.

Therefore, 63 convalescent COVID-19 patients (32.6% of the screened patients, 47 males, mean age 59.82 years, age range 39–65 years) were considered for the final analysis.

### 3.1. Description of the Sample

The demographic and clinical characteristics of the selected patients are shown in Table 1. In brief, the study sample included Caucasian middle-aged patients with a recent history of severe COVID-19 (63.4%). Most patients (77.7%) were transferred from an acute care setting to the rehabilitation setting after hospitalization with a mean length of stay of 25.8 days (SD = 16.5), while 14 (22.3%) came from a home care setting. Five patients (7.9%) were hospitalized in the Intensive Care Unit for a mean time of 33.6 days (range 16–50). The mean time between negativizing of swab and admission to the Rehabilitation Unit was of 38.2 days (range 10–54 days). The most frequent symptoms during the acute phase were persistent insomnia, fever and dyspnoea. Clinical and demographical characteristics of the sample as function of sex are shown in Appendix A.

### 3.2. Psychological Assessment

Participants’ scores on the depression scale of HADS revealed the presence of clinically relevant depressive symptoms (score > 8) in 48/63 patients (76.2%), whereas 11 (17.5%) showed severe depressive symptoms (score > 11). Scores on the HADS-A scale identified high levels of anxiety (score > 8) in 35/63 patients (55.5%) and severe symptoms of anxiety (score > 11) in 26 of them (41.3%). STAI Y2 revealed the presence of moderate trait anxiety (score > 50) in six patients (9.5% of the total sample). No one of these patients had a documented clinical history of depressive or anxiety disorders or episodes prior to SARS-CoV-2 infection. According to current clinical criteria, 28 of the 63 patients (44.4%) presented clinically relevant symptoms of PTSD with an IES-R score >33.

The analysis of internal consistency of self-reported measure showed a Cronbach alpha of 0.82 for HADS-D, a Cronbach alpha of 0.87 for HADS-A, a Cronbach alpha of 0.92 for STAI-Y2 and a Cronbach alpha of 0.85 for the IES-R. The overall trend in the item-total correlations was also high, reflecting the strong relation between single items and a total score (range 0.64–0.87).

### 3.3. Neuropsychological Assessment

In reference to Italian normative data, 11/63 (17.4%) patients showed ‘pathological’ scores (ES = 0) on one neuropsychological test, 5/63 (7.9%) achieved ‘pathological’ scores on 2 tests, and 6/63 (9.5%) had defective scores on more than two tests. Moreover, 26/63 patients (41.2%) showed ‘borderline’ scores (ES = 1) on one test and 12/63 (19.1%) had borderline scores on two or more tests. The number of tests on which the patients obtained ‘pathological’ or ‘borderline’ scores is reported in Table 2. No patients had ‘pathological’ scores (ES = 0) on MoCA or on FAB.

In the verbal fluency task, 21/63 (33.3%) patients produced more than five perseverations; in TMT-Part B 26/63 (41.2%) patients lost the sequence, thus needing the examiner’s help. Raw scores of neuropsychological tests are shown in Appendix A.

On the basis of their neuropsychological scores, 28 patients (44.4% of the sample) were classified as RCE. RCE group did not significantly differ from the NCE group for demographic and most clinical variables, although they tended to show longer duration of bed stay and higher levels of D-Dimer (Table 3). Moreover, the RCE group had significantly higher anxiety scores (U = 322.00) and also tended to show lower scores of EuroQOL (Table 3). Interestingly, the analysis of the distribution of categorical variables (Table 4) showed that RCE more frequently showed alteration of BP circadian rhythm (significantly higher proportion of non-dippers), had experienced a more severe illness as classified by WHO criteria, experienced more clinically relevant symptoms of PTSD (Chi-square = 5.403, df 1, *p* = 0.02), and more frequently complained of cognitive impairments in daily life activities such as poor concentration, difficulty in retrieving words during speech, and reduced ability to remember and to learn new information.

The association between cognitive impairment and alteration of BP circadian rhythm was strikingly confirmed by the finding that within the RCE group all 11 patients who had achieved two or more ‘pathological’ scores (ES = 0) at neuropsychological tests were classified as non-dippers. Correlations between measure of clinical, functional and cognitive status are shown in Appendix A.

## 4. Discussion

The present study investigated the neuropsychological consequences of moderate to severe SARS-CoV-2 infection in patients without relevant premorbid diseases, admitted to a multidisciplinary rehabilitation setting. About half of our sample referred cognitive difficulties consisting of reduced memory, lack of concentration and frequent failures in word retrieval.

The standardized neuropsychological assessment revealed that about one-third of the examined patients had two or more scores in the impaired range of normative samples (ES = 0, ‘pathological’ scores as defined in Italian normative studies) or one ‘pathological’ score and one or more ‘borderline’ scores (ES = 1), thus demonstrating a reduced efficiency in the domains of spatial and verbal long-term memory and executive functions. It is interesting that no patients classified in this RCE group showed ‘pathological’ scores on screening measures of global cognitive efficiency (MoCA) or of frontal functioning (FAB). These findings might suggest that screening cognitive tools might not fully capture the cognitive decline related to COVID-19.

Patients with cognitive impairment showed significantly higher levels of anxiety, and more frequently showed clinically relevant symptoms of PTSD and complained of cognitive difficulties in daily activities, with a trend to experience a lower self-perceived quality of life. Such results would thus extend previous studies on COVID-19 patients affected by reduced cognitive impairment [9,12,14].

As highlighted in previous studies [9,14], our sample of post-COVID-19 patients presented high levels of psychological distress and anxiety symptoms. Indeed, the dramatic experience of hospitalization in COVID-19 Units and the severity of symptoms had a strong impact on physical and psychological health.

In such studies, COVID-19 has been found to be associated with definite consequences on cognitive and psychological conditions, and cognitive impairments have been found to be associated with hypoxia.

Here we did not find a direct and significant association of cognitive impairment with hypoxia, but this negative finding was likely related to the fact that here we enrolled patients with high severity of lung functional alterations (the percentage of hypoxic patients was very high). Indeed, we observed that patients of the RCE group tended to show longer duration of bed stay and higher levels of D-Dimer, and had experienced a more severe illness as classified by WHO criteria. While previous studies have highlighted the association between neurological symptoms and pulmonary function parameters [9], none of them explored the relationship between cognitive impairment and alterations in BP circadian rhythm. Here we observed that a relevant proportion of post-COVID-19 patients with ‘pathological’ scores on neuropsychological tests were non-dippers (patients with nocturnal BP fall < 10%). An alteration of the autonomic nervous system has already been associated with COVID-19 severity, and heart rate variability (HRV) has been indicated as a non-invasive predictor for clinical outcomes in this setting [49]. Similarly to heart rate, control of blood pressure involves both the central and the autonomous nervous system, as well as baroreceptors, hormones, and vascular structures [50]. Our documented alterations of BP circadian rhythm in a significant proportion of COVID-19 patients are in line with the previous evidence on HRV, thus confirming the presence of a dysautonomic disorder following COVID-19 infection [51]. The clinical relevance of this finding can be better understood if we also consider the exclusion of patients with relevant comorbid diseases, including previous cardiovascular diseases and some of the main cardiovascular risk factors. In this regard, we obtained the first empirical evidence supporting the idea that the presence of an altered autonomic function after COVID-19 disease may be associated with measurable impairments in selected cognitive domains in patients without relevant premorbid conditions.

This study has several limitations. First, we based our observations on a relatively small sample size with homogeneous clinical and demographic conditions. The low number of female patients did not allow us to control for the possible influence of sex. Moreover, our study design did not allow us to explore the potential impact of some clinical variables on the development of cognitive impairment after COVID-19. We could not exclude the effect of fluctuations of cognitive performance, depending on mood and medications, particularly for patients who achieved ‘borderline’ scores (ES = 1). Furthermore, sleep, which influences cognitive functioning, was not investigated [52]. For these reasons, we used a conservative approach and classified in the RCE group only patients with two or more pathological scores (ES = 0) [44]. Moreover, to reduce the possible effect of medications on cognitive performances patients were assessed at admission before starting any pharmacological treatment for anxiety and depression, potentially affecting cognition. The lack of a follow-up assessment to evaluate the progression of such reduced cognitive efficiency is another potential limitation. Last, we should consider the lack of measures of cerebral metabolic functioning by means of neuroimaging assessment. Notwithstanding such limitations, our study appears to be of clinical interest, since it emphasizes the association of autonomic disorders with the development of cognitive difficulties. Moreover, differently from previous literature, this study was not limited to the use of screening tools to assess cognitive global impairment, which might be not sensitive enough to capture cognitive decline, but adopted standardized tests for specific domains. Finally, the inclusion of only post-COVID-19 patients without premorbid diseases allowed us to exclude the possible influence of variables that have a well-known effect on cognitive functioning.

## 5. Conclusions

Our findings suggest an association between reduced cognitive efficiency and autonomic disorder in post-COVID-19 patients without premorbid diseases potentially affecting cognition. In particular, we observed that altered BP circadian rhythm could be frequently associated with the presence of cognitive impairment. Future studies with larger sample size and more specific measures of microvascular cerebral functioning could better clarify the mechanisms underlying the cognitive impairment in convalescent COVID-19 patients.

## Figures and Tables

**Table 1 healthcare-10-00480-t001:** Demographic and clinical characteristics of the study sample (*n* = 63).

Characteristics of the Sample	
Sex (F/M), *n* (ratio)	16/47 (1:2.9)
Age (years)	59.82 ± 10.78 (39–65)
Severe pneumonia	40 (63.4%)
Duration of hospital stay (days)	20.17 ± 16.79 (15–90)
Mechanical ventilation	54 (85.7%)
O2 high fluxes	19 (30.1%)
Provenance	
Sub-intensive care unit	25 (39.7%)
Intensive Care Unit	5 (7.9%)
COVID-19 Unit	19 (30.2%)
Home	14 (22.2%)
Clinical features at admission	
PaO2	75.09 ± 15.25 (48–100)
PaCO2	36.08 ± 4.06 (25–88)
FEV1%	78.88 ± 20.99 (34–119)
FVC%	75.59 ± 20.25 (35–113)
FEV1/FVC	83.33 ± 6.89 (62–94)
DLCO%	14.67 ± 8.02 (25–110)
DLCO/VA%	80.46 ± 21.35 (21–109)
Dippers/Non-Dippers, *n*	26/37

Note. PaO2: partial pressure of oxygen in arterial blood; PaCO2: partial pressure of carbon dioxide; FEV1: forced expiratory volume in 1 s; FVC: forced vital capacity; DLCO: diffusion lung capacity for carbon monoxide; VA: alveolar volume. Dippers mean patients with nocturnal fall in blood pressure ≥10%, non-dippers mean patients with nocturnal fall in blood pressure < 10%. Data are expressed as numbers (percentage) or means ± standard deviations (and range).

**Table 2 healthcare-10-00480-t002:** Number of patients with impaired or borderline scores on neuropsychological tests.

	Corsi Span	Corsi-SSL	TMT	RAVLT-IR	RAVLT-DR	Verbal Fluency
‘Pathological’ Scores (ES = 0)	6 (9.5%)	14 (22.2%)	4 (6.3%)	6 (9.5%)	10 (15.8%)	9 (14.2%)
‘Borderline’ Scores (ES = 1)	12 (19%)	8 (12.7%)	10 (15.8%)	8 (12.7%)	6 (9.5%)	8 (12.7%)

Note: Corsi SSL: Supra-span learning on Corsi’s Test; TMT: Trail Making Test part B-A; RAVLT-IR: Immediate recall on Rey’s Auditory Verbal; RAVLT-DR: Delayed recall on Rey’s Auditory Verbal Learning Test.

**Table 3 healthcare-10-00480-t003:** Comparisons of demographical and clinical measures between patients with reduced cognitive efficiency (RCE) and with normal cognitive efficiency (NCE).

	RCE (*n* = 28)	NCE (*n* = 35)	*p*
Sex, (F/M)	7/21	9/26	0.948
Age	58.78 (10.61)	58.78 (11.16)	0.64
Education (years)	11.46 (3.67)	12.73 (3.41)	0.272
Disease duration (days)	51.37 (34.22)	47.75 (23.54)	0.739
Duration of hospital stay (days)	25.10 (19.13)	25.54 (19.38)	0.71
Duration of bed stay (days)	23.79 (17.96)	21.97 (19.3)	0.034
D-Dimer (ng/mL)	832.14 (743.253)	427.32 (229.62)	0.037
CRP (mg/L)	10.95 (13.20)	8.67 (21.18)	0.128
PaO2 (mmHg)	72.67 (14.97)	77.53 (15.20)	0.192
PaCO2 (mmHg)	35.92 (4.72)	37.74 (9.13)	0.726
FEV1 (% predicted)	75.42 (22.80)	81.66 (20.37)	0.316
FVC (% predicted)	70.47 (24.13)	78.31 (17.68)	0.383
FEV1/FVC	83.41 (6.32)	83.28 (7.28)	0.752
DLCO (% predicted)	48.45 (19.71)	63.69 (21.09)	0.055
DLCO/VA (% predicted)	64.50 (32.40)	85.78 (13.71)	0.119
Mean daytime SBP (mmHg)	123.58 (14.04)	120.7 (14.78)	0.825
Mean daytime DBP (mmHg)	79.69 (10.03)	80.03 (10.3)	0.891
Mean nighttime SBP (mmHg)	120.88 (14.77)	112.91 (16.5)	0.062
Mean nighttime DBP (mmHg)	73.85 (9.69)	73.27 (11.37)	0.742
HADS_Anxiety score	10.39 (3.5)	8.25 (2.78)	0.017
HADS_Depression score	12.61 (4.48)	11.56 (4.01)	0.193
STAI-Trait score	40.88 (9.86)	39.34 (8.66)	0.281
Barthel score	69.57 (28.78)	78.06 (22.31)	0.345
EQ-5D Visual Analog Scale	64.03 (12.68)	65.37 (18.17)	0.024

Note. CRP: C-reactive protein; PaO2: partial pressure of oxygen in arterial blood; PaCO2: partial pressure of carbon dioxide; FEV1: forced expiratory volume in 1 s; FVC: forced vital capacity; DLCO: diffusion lung capacity for carbon monoxide; VA: alveolar volume; SBP: systolic blood pressure; DBP: diastolic blood pressure; HADS: Hospital Anxiety and Depression Scale; STAI: State trait Anxiety Questionnaire (trait anxiety); EQ-5D: Euro QoL. Data are expressed as numbers (percentage) or means ± standard deviations (and range). Means were compared by non-parametric Mann-Whitney U test and frequencies by non-parametric Chi-square. Significant Bonferroni-corrected P values are printed in bold; P values approaching Bonferroni-corrected significance threshold are printed in cursive.

**Table 4 healthcare-10-00480-t004:** Distribution of patients as a function of clinical variables and presence of reduced cognitive efficiency.

	RCE	Non-RCE	Totals	Analysis of Frequencies
Hypoxia	13	18	31	
Absence of Hypoxia	15	17	32	Chi-square = 0.156; df = 1; *p* = 0.693
Reduced lung function	17	19	36	
Normal lung function	11	16	27	Chi-square = 0.263; df = 1; *p* = 0.608
Dippers	7	19	26	
Non-Dippers	21	16	37	Chi-square = 5.504; df = 1; ***p* = 0.01**
WHO 3	6	17	23	
WHO 4	22	18	40	Chi-square = 4.944; df = 1; ***p* = 0.02**
PTSD	17	11	28	
No PTSD	11	24	35	Chi-square = 5.403; df = 1; ***p* = 0.02**
Cognitive complaints	19	10	29	
No cognitive complaints	9	25	34	Chi-square = 9.664; df = 1; ***p* = 0.002**

Note. Hypoxia (i.e., PaO2 < 80% predicted); Reduced lung function (i.e., FEV1 < 80% predicted); Dippers (nocturnal BP fall ≥ 10%), Non-dippers (nocturnal BP fall < 10%). *p* values approaching the significance threshold are printed in bold.

## Data Availability

The data supporting the findings of this study are available from the corresponding authors upon reasonable request.

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
