# Peer review of "Cognitive Impairment in Convalescent COVID-19 Patients Undergoing Multidisciplinary Rehabilitation: The Association with the Clinical and Functional Status"

_healthcare, 2022, doi:10.3390/healthcare10030480_

Round 1

Reviewer 1 Report

Overall, it's well written. The authors used adequate methodology and included enough details in different sections with summary tables. Below are minor comments for the authors to consider.

Supplemental Table 2 shows Clinical and demographical characteristics of the sample as a function of sex. It is also better to compare RCE and NCE by sex. Table 3 does not have sex.

I also would try to nuance the conclusion in the abstract. Low sample size is a major limitation. Future research with large samples are needed to provide valid conclusions.

Author Response

We thank the reviewer for these advices that allowed us to improve our paper.

REVIEWER: Supplemental Table 2 shows Clinical and demographical characteristics of the sample as a function of sex. It is also better to compare RCE and NCE by sex. Table 3 does not have sex.

RESPONSE: we added the comparison between RCE and NCE by sex in the Table 3.

REVIEWER: I also would try to nuance the conclusion in the abstract. Low sample size is a major limitation. Future research with large samples are needed to provide valid conclusions.

RESPONSE: we nuanced the conclusions by adding the sentence: “Future research with large samples are needed to provide valid conclusions.”

Reviewer 2 Report

1 Abstract

When possible, choose to use active voice “We consecutively screened for inclusion convalescent COVID-19 patients referring to a post-acute care facility for pulmonary rehabilitation”

  2 Introduction

In this sentence “A number of neurological manifestations (e.g., myalgia, headache, dizziness, hyposmia, hypogeusia, ataxia-encephalitis, stroke, seizures) are considered both a direct effect of the virus infection on the central nervous system and the consequence of immune-mediated reactions during coronavirus disease 2019 (COVID-19) [1, 2].”

Please add “sleep problems” and this reference:

Partinen M et al. (2021). Sleep and daytime problems during the COVID-19 pandemic and effects of coronavirus infection, confinement and financial suffering: a multinational survey using a harmonised questionnaire. BMJ Open 11:e050672. doi:10.1136/bmjopen-2021-050672

In this sentence “ Several studies have also investigated occurrence of cognitive dys-
functions after COVID-19 by means of screening tools or computerized batteries,
showing that cognitive domains, such as memory, attention and speed pro-
cessing, were mainly impaired [9], both in the post-acute phase [7, 10]”

Please include “nightmares” and this reference:

Scarpelli et al. (2022). Nightmares in people with COVID-19: did coronavirus infect our dreams? Nature and Science of Sleep 2022:14 93–108

These two paragraphs may be better togheter:

The effect of the inflammatory response [7] or the psychological outcomes after the hospitalization experience have also been called into question [10].
Since COVID-19 can be considered as a systemic illness, it increases the risk of several non-pulmonary complications such as acute myocardial injury, renal failure or thromboembolic events     Methods   Avoid too much passive voice, like in this paragraph: Circadian blood pressure variations were measured by means of a blood pressure 24-hour Holter monitor. Daytime and nighttime BP were computed as the mean value during daytime and nighttime, respectively. Participants were required to carry out their normal activities during the monitoring period, with the only exception of keeping their arms still at the time of each BP reading. Moreover, patients were asked to record their activities during each measurement.”     Discussion   Maybe is better to put these paragraphs togheter in only one: “ As highlighted in previous studies [7, 12], our sample of post-COVID-19 patients presented high levels of psychological distress and anxiety symptoms. Indeed, the dramatic experience of hospitalization in COVID-19 Units and the severity of symptoms had a strong impact on physical and psychological health.   In such studies, COVID-19 has been found to be associated with definite consequences on cognitive and psychological conditions, and cognitive impairments have been found to be associated with hypoxia.”
In the Limitations session, please say that sleep - which influences cognition - was not investigated. Cite this reference:    Merikanto et al. (2021). Disturbances in sleep, circadian rhythms and daytime functioning in relation to coronavirus infection and Long-COVID – A multinational ICOSS study. Journal of Sleep Research, e13542.

Author Response

We thank the reviewer for advices that allowed us to improve our paper. Changes are highlighted in wellow in the text.

1 Abstract

When possible, choose to use active voice “We consecutively screened for inclusion convalescent COVID-19 patients referring to a post-acute care facility for pulmonary rehabilitation”

We changed the sentence in the abstract according to your suggestion.

 REVIEWER: 2 Introduction

In this sentence “A number of neurological manifestations (e.g., myalgia, headache, dizziness, hyposmia, hypogeusia, ataxia-encephalitis, stroke, seizures) are considered both a direct effect of the virus infection on the central nervous system and the consequence of immune-mediated reactions during coronavirus disease 2019 (COVID-19) [1, 2].”

Please add “sleep problems” and this reference:

Partinen M et al. (2021). Sleep and daytime problems during the COVID-19 pandemic and effects of coronavirus infection, confinement and financial suffering: a multinational survey using a harmonised questionnaire. BMJ Open 11:e050672. doi:10.1136/bmjopen-2021-050672

RESPONSE: we added both “sleep problems” and the related reference in the text.

In this sentence “ Several studies have also investigated occurrence of cognitive dysfunctions after COVID-19 by means of screening tools or computerized batteries, showing that cognitive domains, such as memory, attention and speed processing, were mainly impaired [9], both in the post-acute phase [7, 10]”

Please include “nightmares” and this reference:

Scarpelli et al. (2022). Nightmares in people with COVID-19: did coronavirus infect our dreams? Nature and Science of Sleep 2022:14 93–108

RESPONSE: we included both “nightmares” and the related reference in the text.

These two paragraphs may be better togheter:

The effect of the inflammatory response [7] or the psychological outcomes after the hospitalization experience have also been called into question [10].
Since COVID-19 can be considered as a systemic illness, it increases the risk of several non-pulmonary complications such as acute myocardial injury, renal failure or thromboembolic events.

RESPONSE: We unified the two paragraphs.    

3 Methods  

Avoid too much passive voice, like in this paragraph: Circadian blood pressure variations were measured by means of a blood pressure 24-hour Holter monitor. Daytime and nighttime BP were computed as the mean value during daytime and nighttime, respectively. Participants were required to carry out their normal activities during the monitoring period, with the only exception of keeping their arms still at the time of each BP reading. Moreover, patients were asked to record their activities during each measurement.”

RESPONSE: we modified sentences to avoid much passive forms.   

4 Discussion  

Maybe is better to put these paragraphs togheter in only one: “ As highlighted in previous studies [7, 12], our sample of post-COVID-19 patients presented high levels of psychological distress and anxiety symptoms. Indeed, the dramatic experience of hospitalization in COVID-19 Units and the severity of symptoms had a strong impact on physical and psychological health.   In such studies, COVID-19 has been found to be associated with definite consequences on cognitive and psychological conditions, and cognitive impairments have been found to be associated with hypoxia.”

RESPONSE: We unified the two paragraphs.

In the Limitations session, please say that sleep - which influences cognition - was not investigated. Cite this reference:    Merikanto et al. (2021). Disturbances in sleep, circadian rhythms and daytime functioning in relation to coronavirus infection and Long-COVID – A multinational ICOSS study. Journal of Sleep Research, e13542.

RESPONSE: We added the sentence “Furthermore, sleep, which influences cognition, was not investigated” in the limitation section reference with relating reference.
